**RESEARCH**

# Microbial genomes from non-human primate gut metagenomes expand the primate-associated bacterial tree of life with over 1000 novel species

Serena Manara[1], Francesco Asnicar[1†], Francesco Beghini[1†], Davide Bazzani[1], Fabio Cumbo[1], Moreno Zolfo[1], Eleonora Nigro[1], Nicolai Karcher[1], Paolo Manghi[1], Marisa Isabell Metzger[1], Edoardo Pasolli[2] and Nicola Segata[1*]

## Abstract

**Background:** Humans have coevolved with microbial communities to establish a mutually advantageous relationship that is still poorly characterized and can provide a better understanding of the human microbiome. Comparative metagenomic analysis of human and non-human primate (NHP) microbiomes offers a promising approach to study this symbiosis. Very few microbial species have been characterized in NHP microbiomes due to their poor representation in the available cataloged microbial diversity, thus limiting the potential of such comparative approaches.

**Results:** We reconstruct over 1000 previously uncharacterized microbial species from 6 available NHP metagenomic cohorts, resulting in an increase of the mappable fraction of metagenomic reads by 600%. These novel species highlight that almost 90% of the microbial diversity associated with NHPs has been overlooked. Comparative analysis of this new catalog of taxa with the collection of over 150,000 genomes from human metagenomes points at a limited species-level overlap, with only 20% of microbial candidate species in NHPs also found in the human microbiome. This overlap occurs mainly between NHPs and non-Westernized human populations and NHPs living in captivity, suggesting that host lifestyle plays a role comparable to host speciation in shaping the primate intestinal microbiome. Several NHP-specific species are phylogenetically related to human-associated microbes, such as Elusimicrobia and *Treponema*, and could be the consequence of host-dependent evolutionary trajectories.

**Conclusions:** The newly reconstructed species greatly expand the microbial diversity associated with NHPs, thus enabling better interrogation of the primate microbiome and empowering in-depth human and non-human comparative and co-diversification studies.

**Keywords:** Metagenomic assembly, Non-human primates microbiome, Microbiome sharing, Host-microbiome coevolution

* Correspondence: nicola.segata@unitn.it
†Francesco Asnicar and Francesco Beghini contributed equally to this work.
[1]CIBIO Department, University of Trento, Trento, Italy
Full list of author information is available at the end of the article

## Background

The human microbiome is a complex ecosystem, consisting of diverse microbial communities that have important functions in host physiology and metabolism [1]. The gut microbiome is influenced by several factors including diet [2], physical activity [3], use of antibiotics [4], and other lifestyle-related conditions. Studies comparing the microbiome of rural and industrialized communities have also shown that dietary and lifestyle changes linked to Westernization have played a pivotal role in the loss of many microbial taxa and in the rise of others [5–14]. Although it is difficult to establish causality and mechanisms for these links [15, 16], recent studies have extended the identifiable members of the human microbiome to now cover > 90% of its overall diversity [11], which is a prerequisite for advancing the understanding of the role of microbes in human physiology and metabolism.

A comprehensive understanding of the current structure of the human microbiome needs to consider the study of how the microbiome has coevolved with humans. Ancient intestinal microbiome samples (i.e., coprolites) can give some insights on the gut microbial composition of pre-industrialized and prehistoric humans and date back to a few thousand years [17–21], but the time-dependent degradation issues of microbial DNA limits the possibility of profiling stool samples predating the neolithic period [22]. Some patterns of co-diversification between humans and their microbiomes can be in principle investigated by comparative and phylogenetic analysis of genomes and metagenomes in non-human primates (NHPs), the closest evolutionary relatives of humans [23]. However, a very substantial fraction of the microbiome in NHPs is currently uncharacterized, and a comprehensive comparative sequence-level analysis against human microbiomes is thus unfeasible.

Recent studies of NHPs uncovered part of their hidden microbial diversity but only very partially contributed to the extension of the genetic blueprint of the microbiome in these hosts. Several 16S rRNA gene amplicon sequencing studies investigated the microbiome composition of NHPs [24–32], and some, including a meta-analysis [33], investigated the overlap and specificity of microbial communities associated with humans and NHPs [34–36]. Yet, because this approach has a limited phylogenetic resolution and lacks functional characterization, many co-diversification aspects cannot be studied. Some studies have also applied shotgun metagenomics on NHP microbiomes [30, 37–41], but all of them employed a reference-based computational profiling approach, which solely allows the identification of the very few known microbial species present in NHPs, disregarding those that have not been characterized yet.

However, because of the advances in metagenomic assembly [42, 43] and its application on large cohorts [11], there is now the possibility to compile a more complete catalog of species and genomes in NHP microbiomes and thus enable accurate co-diversification and comparative analyses.

In this study, we meta-analyzed 203 available shotgun-sequenced NHPs metagenomes and performed a large-scale assembly-based analysis uncovering over 1000 yet-to-be-described species associated with NHP hosts, improving NHP gut metagenomes mappability by over 600%. We compared the newly established catalog of NHP-associated species in the context of a large-scale human microbiome assembly project [11] to expose the overlap and divergence between the NHP and human gut microbiome. We showed that captive NHPs harbor microbial species and strains more similar to the human ones compared to wild NHPs and that the extent of microbiome overlap is strongly lifestyle-dependent. Through comparative microbiome analysis, we thus describe the loss of biodiversity from wild to captive NHP that mimics that from non-Westernized to Westernized human populations.

## Results and discussion

To investigate the extent to which the composition of the gut microbiome overlaps across different primates for both known and currently uncharacterized microbes, we meta-analyzed a large set of gut microbiomes from humans and non-human primates (NHPs) that are publically available. Six datasets were considered for NHPs [30, 37–41] spanning 22 host species from 14 different countries in 5 continents (Additional file 1: Table S1 and Additional file 2: Figure S1), totaling 203 metagenomic samples that we retrieved and curated for this work. Microbiome samples from adult human healthy individuals were retrieved from 47 datasets considered in a recent meta-analysis [11] on 9428 human gut metagenomes and used as a comparative resource. Human samples include both Westernized and non-Westernized populations from different countries, whereas NHP datasets cover 4 primate clades, including Old and New World monkeys, apes, and lemurs (Additional file 1: Table S1, Fig. 1a). Two datasets (LiX_2018 and SrivathsanA_2015) surveyed NHPs in captivity, which were fed a specific human-like diet [39] or a diet similar to the one of wild NHPs [38], respectively.

### The newly metagenome-assembled genomes greatly increase the mappable diversity of NHP microbiomes

Reference-based taxonomic profiling of all the 203 samples (see the "Methods" section and Additional file 3: Table S2) confirmed that a very large fraction of NHP metagenomes remains unmapped and uncharacterized (average estimated mapped reads 2.1% ± 3.64% st.dev.,

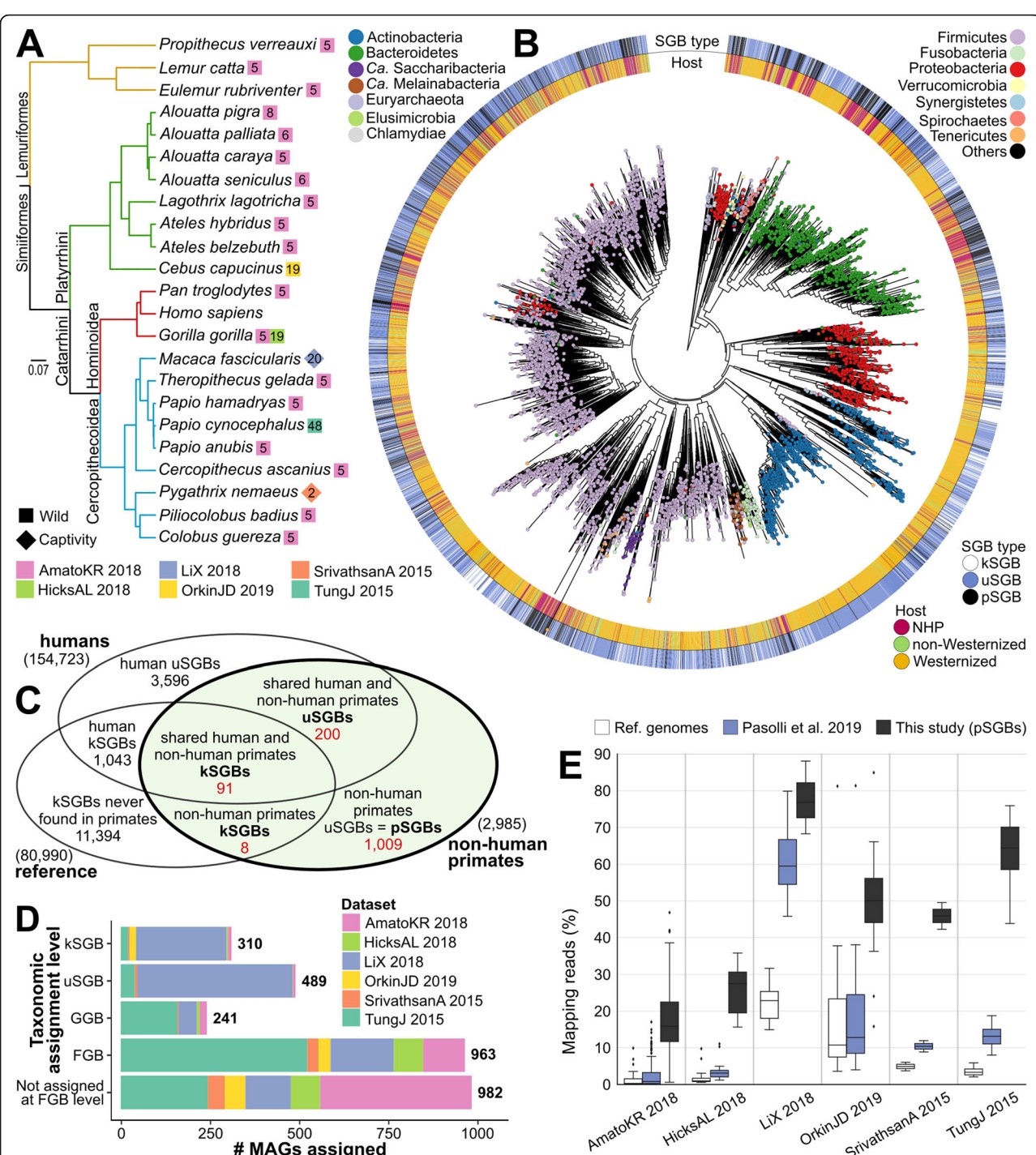

**Fig. 1** The expanded set of microbial genomes and species from the gut microbiomes of NHPs. **a** Phylogenetic tree of the primate species considered in this study (adapted from [44]), reporting the dataset and number of samples per species. **b** Microbial phylogeny of the 4930 species-level genome bins (SGBs, using single representative genomes, see the "Methods" section) and the 1009 SGBs that are specific to NHPs and newly retrieved in this study. **c** Overlap between the sets of SGBs reconstructed from NHP metagenomes and at least 1 reference microbial genome (kSGBs), between SGBs reconstructed both from NHP and human metagenomes but lacking a reference genome (uSGBs), and identification of newly assembled SGBs from NHPs metagenomes only (pSGBs). **d** Fraction of MAGs assigned to clades at different taxonomic levels; samples unassigned at the species level (kSGB or uSGB) could be assigned to known genus-level genome bins (GGBs) or family-level genome bins (FGBs), or remained unassigned at the family level (not assigned at FGB level). **e** Statistics of NHP metagenomic read mappability before and after the addition of MAGs from human and NHP metagenomes. We observed an average increase of 612% with respect to the reference genomes alone and 206% with respect to the catalog of human MAGs

Additional file [4]: Table S3). This points at the paucity of microbial genomes representative for members of the gut microbiome of NHPs, which greatly overcome the unexplored diversity still hidden in human microbiomes [45]. We thus employed an assembly-based approach we previously proposed and validated elsewhere [11] (see the "Methods" section) and that was also recently cross-checked with other similar efforts [46] to reconstruct microbial genomes de novo in the whole set of available NHP metagenomic samples. After single-sample assembly and contig binning of the 203 NHP metagenomes considered, we retrieved a total of 2985 metagenome-assembled genomes (MAGs) (Additional file [5]: Table S4) that exceeded the threshold for being considered of medium quality (completeness > 50% and contamination < 5%) according to recent guidelines [47]. A large fraction of these genomes (34.6%) could additionally be considered of high quality (completeness > 90% and contamination < 5%) and provide the basis for assessing the diversity of NHP microbiomes

Functional annotation of all MAGs (see the "Methods" section [48]) showed low levels of functional characterization in NHPs, with only $1049 \pm 482$ UniRef50 ($61.9\% \pm 17.3\%$ st.dev. of predicted proteins) assigned per MAG, in contrast with the $1426 \pm 591$ ($77.3\% \pm 14.6\%$ st.dev. of predicted proteins) assigned to MAGs from non-Westernized human samples and $1840 \pm 847$ ($83.7\% \pm 12.6\%$ st.dev. of predicted proteins) assigned to those obtained from Westernized human populations. Comparative functional analysis between human and NHP strains was hindered by the low level of overlap between species-level genome bins (SGBs; i.e., clusters of MAGs spanning 5% genetic diversity, see the "Methods" section) retrieved from human and NHP metagenomes, with only 8 SGBs being present in at least 10 human and 10 NHP microbiomes. Statistical analysis on the functional annotations of these shared SGBs showed 150 KEGG Orthologies (KOs) significantly associated with NHP strains and 166 KOs associated with human strains (Fisher's test FDR-corrected *p* values < 0.05, Additional file [6]: Table S5). Among the functions associated with NHP strains, we found different genes involved in the degradation of sugars like cellobiose (K00702, K02761) and maltose (K16211, K01232), and among those associated with human ones, genes encoding for the degradation of different antibiotic compounds, including penicillin and vancomycin (K01710, K02563, K07260, K07259), which is consistent with the exposure of humans but not NHPs to antibiotics.

We first mapped the 2985 obtained MAGs against the previously described SGBs that recapitulate the > 150,000 MAGs from the human microbiome and the > 80,000 reference microbial genomes from public repositories. In total, 310 MAGs (10.39%) fell into 99 SGBs

containing at least 1 known reference genome (called kSGBs), whereas 489 (16.38%) belonged to 200 unknown species (called uSGBs) lacking reference genomes but previously identified in the human microbiome (Fig. 1c and Table 1). The large majority of the MAGs remained however unassigned, with 2186 MAGs (73.23%) showing > 5% genetic distance to any SGB and 1903 MAGs (63.75%) showing > 10% genetic distance. These completely unknown MAGs firstly reconstructed in this work from NHPs' gut metagenomes were de novo clustered into 1009 NHP-specific SGBs (here defined as primate SGBs or pSGBs) with the same procedure that defines SGBs at 5% genetic diversity we previously employed and validated [11] (Fig. 1c and Table 1). Overall, NHP microbiomes comprised 1308 SGBs covering 22 phyla (Fig. 1b) that expanded the known NHP microbiome diversity with new candidate species mostly expanding the Firmicutes, Bacteroidetes, Euryarchaeota, and Elusimicrobia phyla. On the contrary, Actinobacteria were generally underrepresented among NHP SGBs (Fig. 1b). Although some species were shared between NHPs and humans, our analysis highlighted extensive microbial diversity specifically associated with primates other than humans.

This expanded set of genomes improved the fraction of metagenomic reads in each metagenome that could be mapped by over 6-folds (612%) with respect to the sole reference genomes available in public repositories (> 80,000, see the "Methods" section) and by 2-folds (206.5%) with respect to the catalog of genomes expanded with the MAGs from over 9500 human metagenomes [11] (Fig. 1e). Overall, the average metagenome mappability reached 38.2%, with however uneven increase across datasets (Fig. 1e). The LiX_2018 dataset of NHPs in captivity reached a mappability of 77.6%, whereas the AmatoKR_2018 dataset of wild NHPs merely reached a 17.4% mappability rate (Fig. 1e). The fact that LiX_2018 was already highly mappable even when using the available reference genomes alone (22.2% w.r.t. 1% of AmatoKR_2018) and that the human

**Table 1** Number and percentage of MAGs assigned to different SGB types in the different datasets

| Dataset | Number (%) of MAGs | | | |
|---|---|---|---|---|
| | SGBs | kSGBs | uSGBs | pSGBs |
| SrivathsanA_2015 | 92 | 4 (4.4%) | 7 (7.6%) | 81 (88%) |
| TungJ_2015 | 985 | 21 (2.1%) | 39 (4%) | 925 (93.9%) |
| AmatoKR_2018 | 578 | 10 (1.7%) | 7 (1.2%) | 561 (97.1) |
| HicksAL_2018 | 177 | 4 (2.2%) | 1 (0.6%) | 172 (97.2%) |
| LiX_2018 | 1043 | 253 (24.3%) | 435 (41.7%) | 355 (34%) |
| OrkinJD_2019 | 110 | 18 (16.4%) | 0 (0%) | 92 (83.6%) |
| Total | 2985 | 310 (10.4%) | 489 (16.4%) | 2186 (73.2%) |

SGB database was responsible for the largest increase in mappability (reaching 60.7%, w.r.t. 3% of AmatoKR_ 2018) further confirms that microbiomes from NHPs in captivity are more similar to human ones (Fig. 1e) than those from wild hosts. Also, the TungJ_2015 dataset reached high mappability levels (63.9%), but this was expected as this is the largest dataset in our meta-analysis (23.6% of the samples considered in this study), with all samples ($n = 48$) from the same host. The AmatoKR_ 2018 cohort, on the contrary, surveyed many different wild hosts ($n = 18$, 95 samples) that are not covered by other datasets and that have therefore a limited sample size, explaining the modest gain in mappability (14.4% with respect to the human catalog). Overall, almost 3000 MAGs provide the basis for a deeper understanding of the composition and structure of the primate's gut microbiome.

## Only few and mostly unexplored gut microbes are in common between humans and NHPs

We first investigated how many of the microbial species identified in NHPs were also detected at least once in the human gut microbiome, finding only about 20% overlap (291 of the 1308 SGBs) between NHP and human gut microbial species. Considering the whole set of SGBs found at least once in human or NHP gut metagenomes, this overlap is further reduced to 5.95%. Many of the species found both in NHPs and humans (200 MAGs, 68%) are currently unexplored species without reference genomes (uSGBs). In addition, very few of the newly recovered MAGs belonged to species previously isolated from NHPs but never found in human microbiome samples. This set of 8 known species includes *Helicobacter macacae*, which can cause chronic colitis in macaques [49, 50], and *Bifidobacterium moukalabense*, whose type strain was originally isolated from *Gorilla gorilla gorilla* samples [51], and we reconstructed from two samples of the same host (Additional file 7: Table S6). The other 6 known species (*Fibrobacter* sp. UWS1, *Caryophanon tenue*, *Staphylococcus nepalensis*, *Staphylococcus cohnii*, *Enterococcus thailandicus*, *Serratia* sp. FGI94) comprise 1 MAG only from our dataset and confirm the paucity of isolated and characterized taxa specifically associated to NHPs.

When looking at the species with previously assigned taxonomic labels, we found a total of 91 species with sequenced representatives (kSGBs) in NHPs that can also be found in the human microbiome. However, many of them (64.65%) are still rather uncharacterized species as they represent sequenced genomes assigned to genus-level clades without an official species name (e.g., with species names labeled as "sp." or "bacterium," Additional file 8: Table S7). Most of such relatively unknown kSGBs were from the *Clostridium* genus (15 kSGBs),

and several others belonged to the *Prevotella* (9) and *Ruminococcus* (6) genera. However, both the 2 most represented human kSGBs assigned to the *Prevotella* genus (13 and 11 MAGs recovered, respectively, Fig. 2a and Additional file 9: Table S8) were retrieved from *Macaca fascicularis* in captivity from the LiX_2018 dataset, consistently with previous the literature [36, 52, 53]. Among those kSGBs with an unambiguously assigned taxonomy, 2 highly prevalent *Treponema* species, *T. berlinense* and *T. succinifaciens*, were reconstructed from 14 and 11 samples, respectively, from different studies and host species (Fig. 2a and Additional file 8: Table S7). These two species were previously found to be enriched in non-Westernized populations [11], with 45 genomes reconstructed from different countries. *T. berlinense* and *T. succinifaciens* may thus represent known taxa that are common to primate hosts but that are under negative selective pressure in modern Westernized lifestyles.

The majority (68.7%) of the 291 species shared between humans and NHPs are SGBs without available reference genomes and taxonomic definition (i.e., uSGBs, Fig. 1c, d). Many of these uSGBs remain unassigned also at higher taxonomic levels, with only 25 of them assigned to known genera and 102 to known families. Overall, more than one third (36.5%) of the uSGBs shared with humans were highly uncharacterized and were left unassigned even at the family level (Additional file 9: Table S8). Among these, 5 out of the 10 most prevalent shared uSGBs (accounting for 61 MAGs in total) were assigned to the Bacteroidetes phylum (Fig. 2a) but remained unassigned at lower taxonomic levels (Additional file 9: Table S8). Even among uSGBs, the *Treponema* genus was highly represented, with 9 genomes reconstructed from different samples of *Papio cynocephalus* from the TungJ_2015 dataset (Additional file 9: Table S8). Common human-NHP taxa thus represent only a small fraction of the primate microbiome, and these taxa generally belong to very poorly characterized taxonomic clades.

## Species overlap between human and NHP microbiomes is heavily lifestyle-dependent

Microbiomes of NHPs in captivity showed reduced numbers of previously unseen microbial diversity (pSGBs) and a larger set of strains from species also found in humans (kSGBs and uSGBs) when compared to wild NHPs. Indeed, eight of the ten most prevalent human-associated SGBs found in at least five NHP samples (Additional file 9: Table S8) were recovered from the LiX_2018 and SrivathsanA_2015 datasets, the only two studies which surveyed the microbiome of NHPs in captivity. Accordingly, a high fraction of genomes reconstructed from the LiX_2018 captive dataset matches previously described species (64.2%), in contrast with an

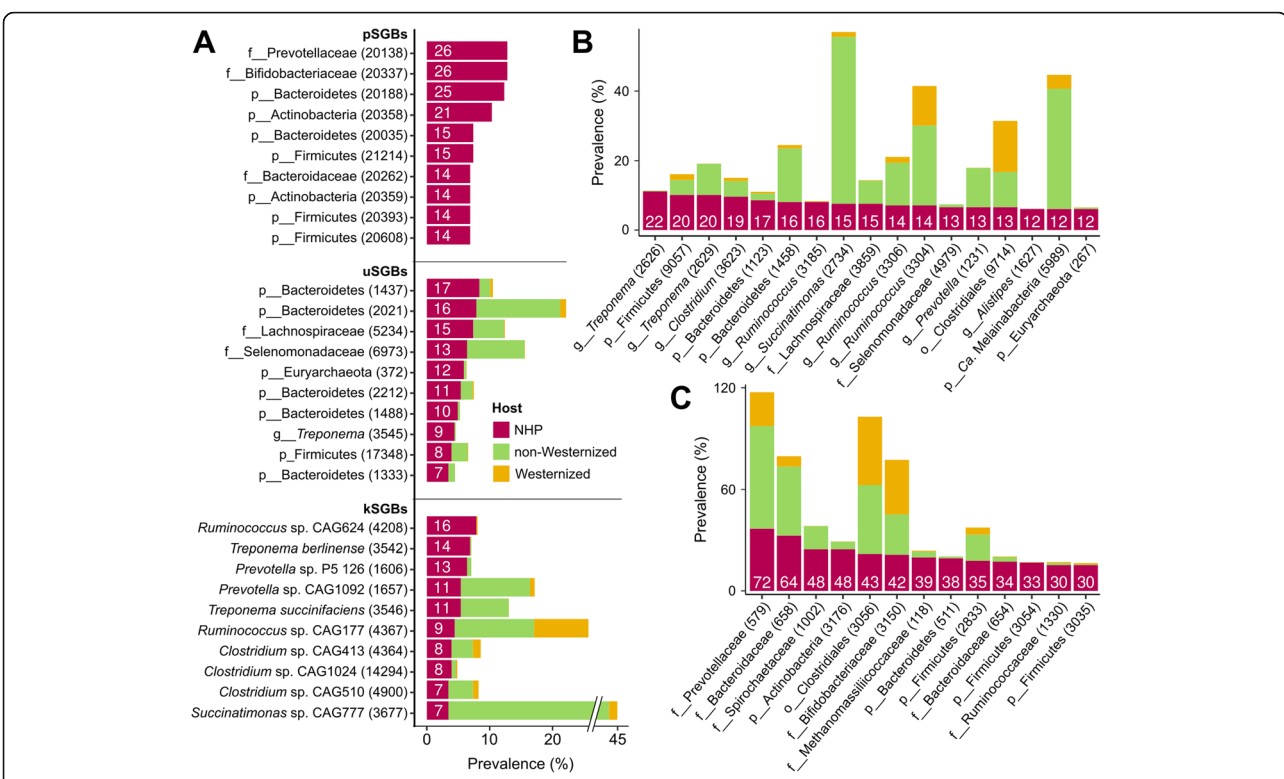

**Fig. 2** Most prevalent NHP genome bins from species level to family level and their prevalence in Westernized and non-Westernized human populations. **a** Most prevalent pSGBs, uSGBs, and kSGBs in NHPs and their prevalence in Westernized and non-Westernized humans. **b** Most prevalent GGBs in NHPs (> 11 NHP samples) and their prevalence in Westernized and non-Westernized humans. **c** Most prevalent FGBs in NHPs (≥ 30 NHP samples) and their prevalence in Westernized and non-Westernized humans. Numbers inside the bars represent the number of NHP samples in which the specific SGB, GGB, or FGB has been found. Full list of SGBs, GGBs, and FGBs is in Additional file 9: Table S8 and Additional file 11: Table S10

average of 7.0% ± 6.0% for the MAGs in wild datasets (Additional file 9: Table S8). Overall, these numbers suggest that the microbiome of captive animals is a rather poor representation of the real diversity of their microbiome in the wild and that exposure of NHPs to the human-associated environment and somehow human-like diet and sanitary procedures can inflate the similarity between human and NHP microbiomes. Nevertheless, a few SGBs were consistently found in both wild and captive NHPs and shared with humans. These ten kSGBs and eight uSGBs mainly belonged to unclassified Firmicutes (*n* = 5) and uncharacterized *Ruminococcus* species (*n* = 4). Among the most prevalent in NHPs, the kSGBs of *Treponema berlinense*, *Succinatimonas* sp., *Escherichia coli*, and *Prevotella* sp. were consistently found in different host species spanning NHPs and humans and thus appear as key players in the primate gut microbiome.

The overlap in microbiome composition between wild NHPs and humans is mostly due to the sharing of SGBs characteristic of microbiomes of non-Westernized rather than Westernized human hosts. This is clear when observing that only 3 SGBs present in NHPs are enriched

in prevalence in stool samples from Westernized populations (Fisher's test, Bonferroni-corrected *p* values < 0.05), in comparison with 41 SGBs enriched in non-Westernized datasets (Fig. 3 and Additional file 10: Table S9). Even for those three SGBs associated with Westernized populations, the average prevalence in Westernized datasets was only 0.42%. The SGB found in NHPs that is most strongly associated with non-Westernized populations is *Succinatimonas* sp. (kSGB 3677, prevalence 41.6% in non-Westernized datasets, 1.3% in Westernized datasets; Fisher's test, Bonferroni-corrected *p* value 2.74E−223, Fig. 3), from a genus able to degrade plant sugars such as D-xylose, a monosaccharide present in hemicellulose and enriched in diets rich in plant products. The broader *Succinatimonas* genus-level cluster also had a prevalence of 48.05% in non-Westernized datasets and of 1.4% in Westernized ones (Fig. 2b), in agreement both with the folivore diet of most NHPs considered here and with previous observations of enriched D-xylose degradation pathways in non-Westernized populations [54]. Overall, the 3 most prevalent genus-level genome bins in NHPs (2 from the *Treponema* genus and 1 from the Firmicutes, all > 10%

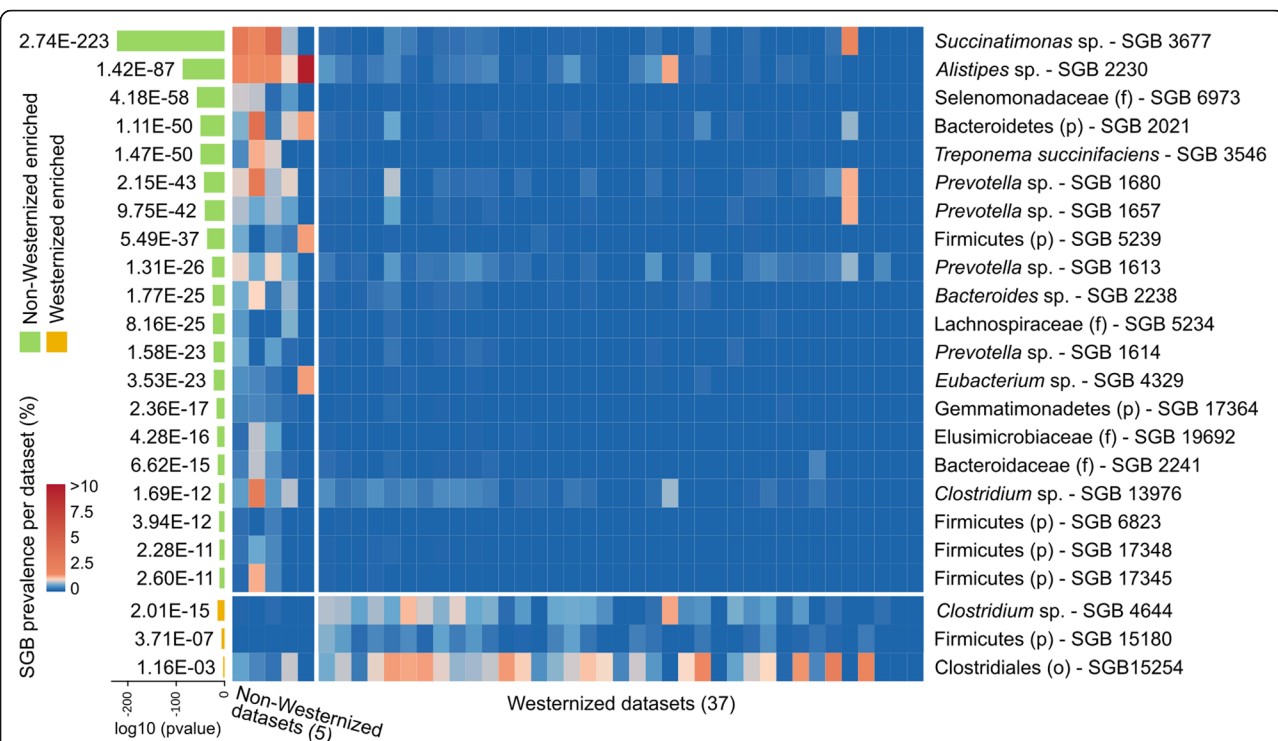

**Fig. 3** Prevalences of the NHP SGBs found in humans differentially present in Westernized or non-Westernized human populations. Association of SGBs found in at least three NHP metagenomes with the gut microbiome of Westernized or non-Westernized populations, together with their prevalence in the different datasets (Fisher's test Bonferroni-corrected *p* values, full results in Additional file 10: Table S9)

prevalence in NHPs) had an average prevalence of 4.5% in non-Westernized and of 0.6% in Westernized populations (Fig. 2b).

At the family-level, many *Prevotella* SGBs are both very prevalent in NHPs and in non-Westernized human populations. The overall *Prevotellaceae* family is the most prevalent in NHPs (36.55%), and its prevalence is even higher in non-Westernized human microbiomes (60.55%), while not reaching 20% in Westernized ones (Fig. 2c). Consistently, 4 out of the 20 SGBs most associated with non-Westernized human populations belonged to the *Prevotella* genus (SGBs 1680, 1657, 1613, 1614, Fig. 3) and were however retrieved only from the LiX_2018 dataset of captive *Macaca fascicularis*. Similarly, the only shared SGB assigned at the species level was *Treponema succinifaciens* (kSGB 3546), which was present in 8.22% of non-Westernized samples and in only 0.02% Westernized microbiomes (Fig. 3 and Additional file 10: Table S9), but all of the samples were from the 2 datasets of NHPs in captivity (LiX_2018 and SrivathsanA_2015), supporting once again the observation that when well-characterized species are found in NHPs, these are usually from captive hosts. The family Spirochaetaceae, to which the genus *Treponema* belongs, was however prevalent also in wild NHPs (24.37%) and non-Westernized samples (13.67%), while being almost

absent in Westernized ones (0.13%, Fig. 2c). These data thus suggest that the level of similarity between human and NHP microbiomes depends not only on the host species but also on lifestyle variables that could be at least partially assessed both in NHPs (wild vs captive animals) and humans (Westernized vs non-Westernized populations).

## Most microbial genomes from NHP metagenomes belong to novel species

More than two thirds (2186) of the MAGs recovered from NHPs (2985) belonged to the 1009 newly defined and previously unexplored SGBs (pSGBs) never found in human microbiomes so far. Some of these pSGBs seem to be key components of the NHP microbiome, with 6 of them (recapitulating 128 MAGs) within the 10 most prevalent SGBs in NHP microbiomes (Fig. 2a and Additional file 9: Table S8). The distribution of pSGBs was however not homogeneous among datasets, with the LiX_2018 dataset being the one with the highest fraction of MAGs assigned to known species (23.5% of the MAGs assigned to kSGBs) and AmatoKR_2018 having 97.23% of the MAGs unassigned at the species level (56.57% unassigned at the family level, Fig. 1d). This again reflects the different composition of the two datasets, with the captive *Macaca fascicularis* of the LiX_

2018 dataset fed with specific human-like diets [39] and the AmatoKR_2018 dataset spanning 18 NHP species living in the wild, which explains its high diversity (Fig. 1a).

Many of the 1009 pSGBs were taxonomically unplaced even at higher taxonomic levels, with only 109 pSGBs assigned to a known microbial genus (10.8%, 241 MAGs, see the "Methods" section) and 386 pSGBs to a known microbial family (38.3%, 963 MAGs, Fig. 1d). The 514 pSGBs (50.9%, 982 MAGs) that remained unassigned may represent new microbial clades above the level of the bacterial families (Fig. 1d). The majority of these pSGBs unassigned even at the genus level or above were placed, based on genome similarity, into the 2 highly abundant human gut microbiome phyla of the Firmicutes (44.2% of the unassigned pSGBs, 514 total MAGs) and Bacteroidetes (30.9% of the unassigned pSGBs, 458 MAGs) with smaller fractions assigned to Proteobacteria (9.7%, 125 MAGs), Actinobacteria (5.5%, 108 MAGs), and Spirochaetes (2.8%, 37 MAGs). Because the dominance of the Bacteroides and Firmicutes phyla is quite consistent among the gut microbiomes of primates, it is thus at the species and genus level that most of the inter-host diversity is occurring, possibly as a consequence of host co-speciation or co-diversification evolutionary dynamics.

To better taxonomically characterize these unassigned pSGBs, we grouped them into clusters spanning a genetic distance consistent with that of known genera and families [11] generating genus-level genome bins (GGBs) and family-level genome bins (FGBs). This resulted in the definition of 760 novel GGBs (73.6% of the total number of GGBs in NHP) and 265 novel FGBs (65.6% of all FGBs in NHP), with an increase of about 6% of the total GGBs and FGBs previously defined on reference genomes and > 154,000 human MAGs. Eight of the 10 most prevalent GGBs in NHP samples were part of this novel set of GGBs and were assigned to Coriobacteriales (36 MAGs), Bacteroidaceae (36 MAGs), and Prevotellaceae (33 MAGs) families. Among the most prevalent, only the 2 *Treponema* GGBs (42 MAGs from NHPs) were known and shared with humans (52 MAGs), mainly from non-Westernized populations (38 MAGs, Fig. 2b and Additional file 11: Table S10). On the contrary, all of the 10 most prevalent families were previously known and shared with humans (Additional file 11: Table S10). In the study of the overall diversity of the primate gut microbiome, it is thus key to consider the new sets of NHP gut microbes defined here that are largely belonging to novel microbial clades.

## Strain-level analysis highlights both host-specific and shared evolutionary trajectories

Despite the low overall degree of microbial sharing between human and non-human hosts at the species level, some bacterial families were common among primate hosts (Fig. 2c) and motivated a deeper phylogenetic analysis of their internal genetic structure. By using a phylogenetic modeling based on 400 single-copy universal markers [55], we reconstructed the phylogeny and the corresponding genetic ordination analysis of the 5 most relevant shared FGBs (Fig. 2c), which included 3 known families (Prevotellaceae, Bacteroidaceae, Spirochaetaceae), and 2 unexplored FGBs assigned to the Actinobacteria phylum and the Clostridiales order. We observed the presence of both intra-family host-specific clusters (Fig. 4a) and clusters comprising genomes spanning human and non-human hosts. The phylogeny of the Bacteroidetes reconstructed to include all of the MAGs and reference genomes for the 10 most prevalent characterized (kSGBs), uncharacterized (uSGBs), and newly reconstructed NHP-specific (pSGBs) species assigned to this phylum (Fig. 4b and Additional file 2: Figure S2) further confirms the presence of closely related sister clades one of which is specific to wild NHPs and the other spanning multiple hosts, including NHPs in captivity. This likely reflects a complex evolutionary pattern in which vertical co-diversification [56, 57], independent niche selection, and between-host species transmission are likely all simultaneously shaping the members of the gut microbiome of primates.

To further investigate the hypothesis of at least occasional paired primate-microbe co-diversification, we selected the taxonomically unassigned FGB 4487, which is the only FGB retrieved in this work that spans 3 out of the 4 main host clades (Lemuriformes, Platyrrhini, Cercopithecoidea, but no Hominoidea), including 15 MAGs reconstructed from 7 wild hosts from 6 countries. The phylogeny of FGB 4487 recapitulated the one of the hosts (Additional file 2: Figure S3), with different same-clade host species from different countries sharing the same SGB (e.g., different *Alouatta* species from 3 different countries sharing pSGB 20386) while being distinct from those found in other clades, thus supporting the hypothesis that host-microbiome co-diversification could have occurred at least for some bacterial clades.

We also analyzed the under-investigated phylum of the Elusimicrobia as species in this clade were already shown to span a wide range of host environments ranging from aquatic sites to termite guts [58] and were recently found relatively prevalent in non-Westernized human populations (15.4% prevalence) while almost absent in Westernized populations (0.31% prevalence) [11]. The phylum was clearly divided into two main clades (Additional file 2: Figure S4), with one including strains mostly from environmental sources or non-mammalian hosts and the other (already reported in Fig. 4c) comprising all the MAGs from humans, NHPs, rumen, and the type strain of *Elusimicrobium minutum* [59]. The

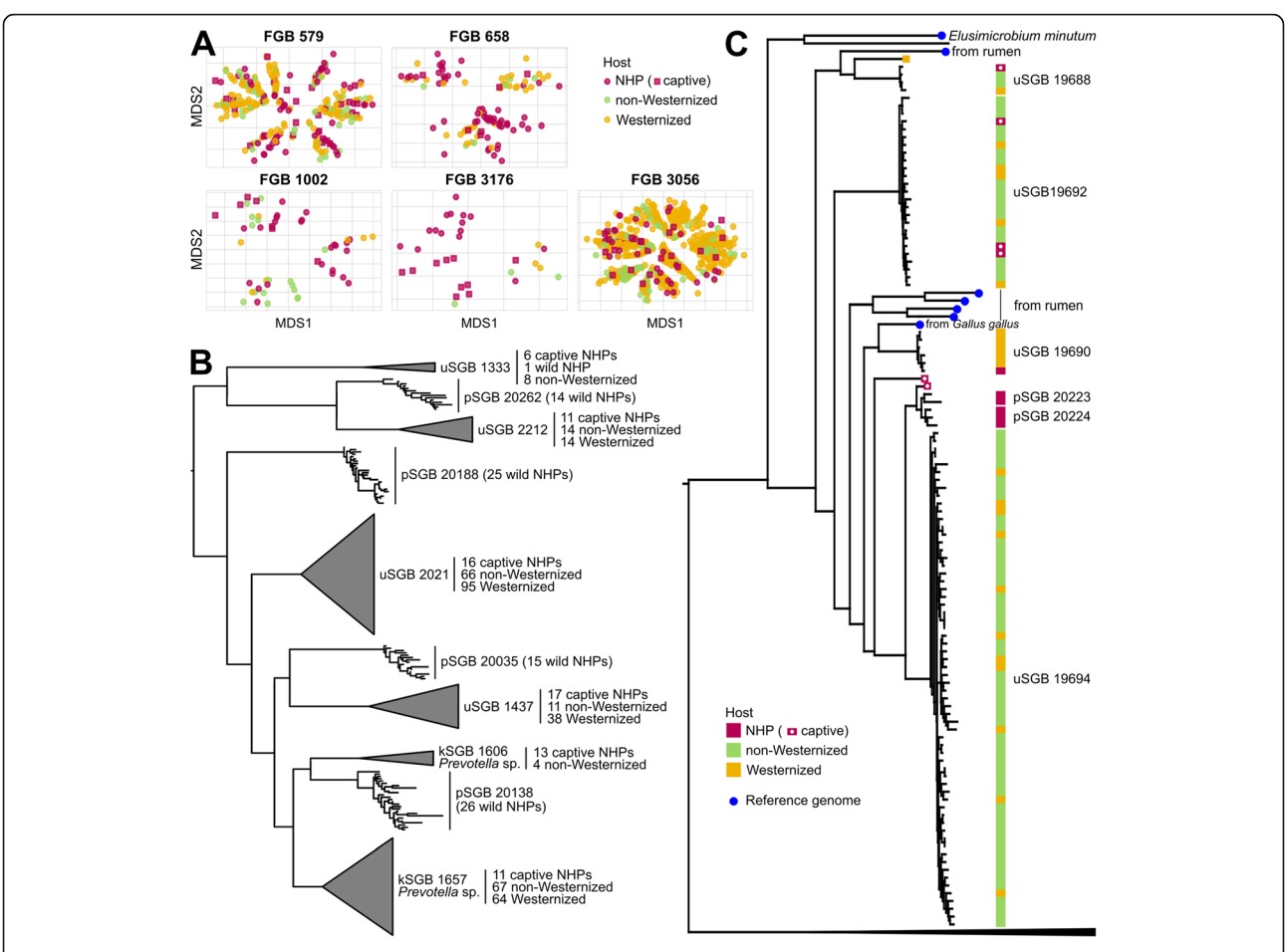

**Fig. 4** Strain-level phylogenetic analysis of relevant microbial clades found both in NHPs and human microbiomes. **a** Ordination analysis using multidimensional scaling (MDS) on intra-FGB phylogenetic distances for the five most prevalent FGBs shared by NHPs and humans (Fig. 2c), showing both host-specific and shared clusters. **b** Phylogenetic tree of the ten most prevalent kSGBs, uSGBs, and pSGBs assigned to the Bacteroidetes phylum reported in Fig. 2a, with MAGs from wild NHPs in separate pSGB subtrees and captive NHPs clustering into SGBs shared with humans (uncollapsed tree in Additional file 2: Figure S2). **c** Phylogenetic tree of the Elusimicrobia phylum, with SGBs specifically associated with wild NHPs and others with humans and captive NHPs (uncollapsed tree in Additional file 2: Figure S4)

genomes from wild NHPs belonged to an unknown SGB detected also in humans (uSGB 19690) and to 2 pSGBs (pSGBs 20223 and 20224) not found in human hosts. These 2 NHP-specific Elusimicrobia are sister clades of a relatively prevalent human-associated SGB (SGB 19694 comprising 64 MAGs from humans, Fig. 4c). Such closely related but host-specific sister clades might again reflect the evolutionary divergence of the hosts, while the presence of Elusimicrobia strains from macaques in captivity inside human-associated SGBs (Fig. 4c) also confirms that these microbes can colonize different primate hosts.

## Closely phylogenetically related *Treponema* species have different host type preferences

The *Treponema* genus contains mostly non-pathogenic species commonly associated with the mammalian intestine and oral cavity [60]. *Treponema* species seem to be

under particular negative selection forces in Westernized populations as multiple studies found them at much higher abundance and prevalence in non-Westernized populations [7, 11, 54, 61, 62], and they were also identified in ancient coprolites [19], and dental calculus of the Iceman mummy [63]. To better study its diversity and host association, we investigated the phylogeny of this genus considering all the genomes from NHPs and humans currently available (Fig. 1b). The 221 total genomes included 27 available reference genomes and 220 MAGs (96 oral and 124 intestinal) spanning 54 *Treponema* SGBs. These genomes are grouped into 34 distinct SGBs previously reconstructed from human metagenomes and 20 pSGBs newly reconstructed and uniquely associated with NHPs.

Phylogenetic analysis (Fig. 5a) highlighted a clear and host-independent separation of oral and stool treponemas that is reflected at the functional level (Fig. 5b), with

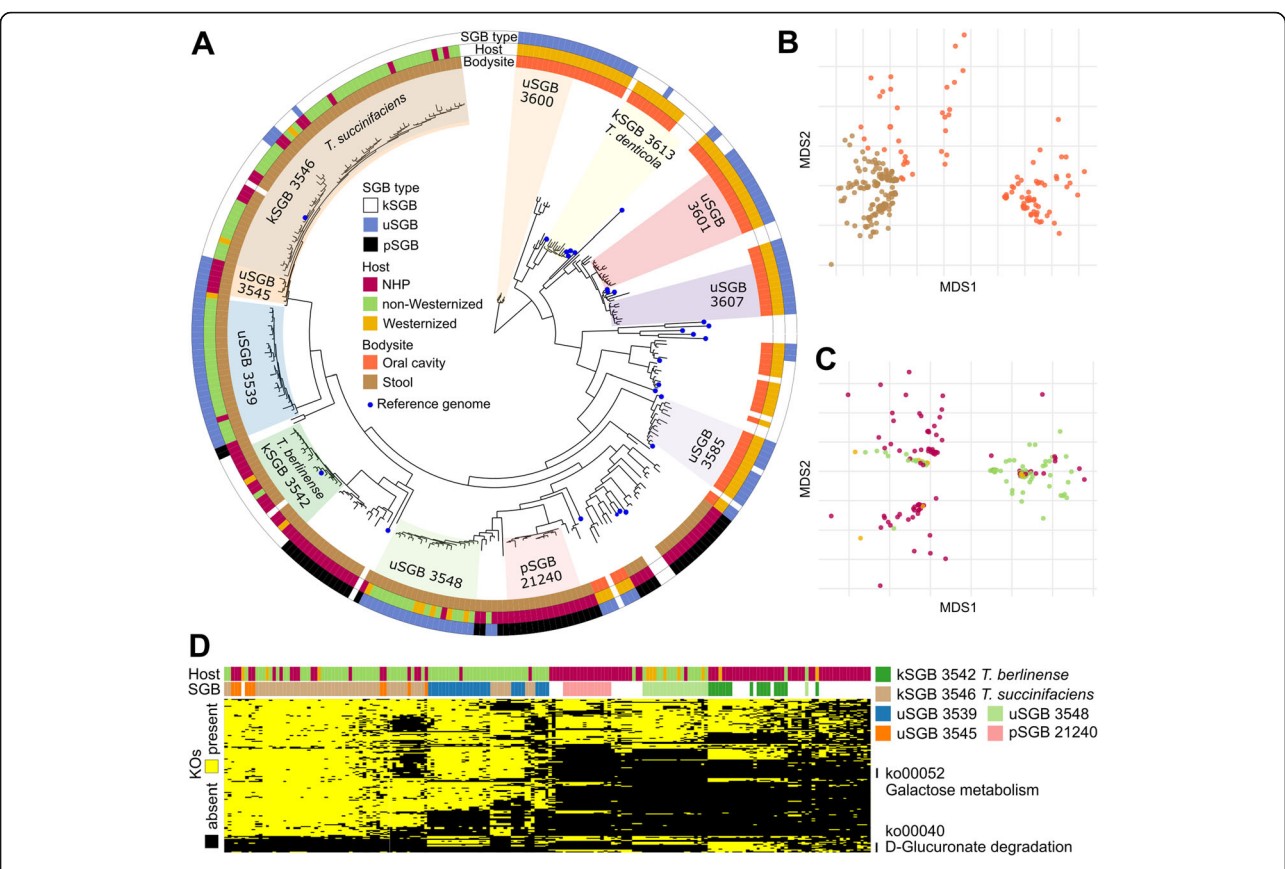

**Fig. 5** The *Treponema* genus is the most prevalent among NHPs. **a** Phylogenetic tree of the *Treponema* genus, showing SGB host specificity and a clear separation between oral and intestinal species (SGB annotation for > 10 genomes). **b** Ordination on functional annotations (UniREf50 clusters) of *Treponema* MAGs colored by body site showing separation of oral and intestinal MAGs at the functional level. **c** Ordination on UniRef50 profiles of *Treponema* MAGs from stool samples only colored by host, showing host-specific functional profiles. **d** Presence/absence profiles of KEGG Orthology families (KOs) in *Treponema* MAGs recovered from stool samples (only KOs related to the metabolism and present in at least 20% and less than 80% of samples are reported)

oral species lacking several pathways encoded by SGBs recovered from stool samples. These included starch and sucrose metabolism, glycerolipid and glycerophospholipid metabolism, methane and sulfur metabolism, folate biosynthesis, and phenylalanine, tyrosine, and tryptophan biosynthesis (Additional file 2: Figure S5), consistently with the nutrients and carbon sources available in the 2 different body sites. Focusing on the intestinal species, the SGBs in this family were quite host-specific, with genomes recovered from different hosts clustering in specific subtrees (Fig. 5a). This is for instance the case of uSGB 3548 and pSGB 21240 that, despite being phylogenetically related, were found only in humans and NHPs, respectively (Additional file 12: Table S11). *Treponema succinifaciens* (kSGB 3546) was instead an exception, as it was possible to reconstruct genomes for this species both from NHP microbiomes (11 MAGs) and (mostly) non-Westernized human stool microbiomes (45 MAGs, Fig. 5a, Additional file 12: Table S11). However, the closely related uSGB 3545 was

recovered only from NHPs (*Papio cynocephalus*) and could represent a species specifically adapted to the gut of these NHPs or the consequence of the host speciation. It is quite striking that only 11 *Treponema* MAGs were available from Westernized stool samples despite the large number of gut metagenomes analyzed for this category (7443 stool samples), whereas the same microbial genus was very prevalent in non-Westernized datasets (13.72% of non-Westernized samples, all but one non-Westernized datasets, Fig. 5a and Additional file 6: Table S5). This raises the hypothesis that *Treponema* species might have been living within the gut of their primate hosts for a long time and have remained with humans in the absence of lifestyle changes associated with urbanization [22].

The host specificity of related *Treponema* species is evident also at the functional level (Fig. 5c) with several microbial pathways characterizing each species. When comparing the functional potential across hosts, we found for example that human strains were enriched for

genes necessary for galactose metabolism (ko00052) and NHPs strains were instead encoding the pathway for the degradation of glucuronate-containing polymers (ko00040), highly present in hemicellulose (Fig. 5d), consistently with the different nutritional regimes of humans and NHPs. *Treponema* species enriched in NHPs were however including a substantially lower number of annotated functions (1312 ± 375 in NHPs w.r.t. 1426 ± 423 UniRef50 in Westernized samples), pointing to the need of future efforts to experimentally characterize the genes in under-investigated NHP species. The *Treponema* genus overall appears to be a key member of the primate-wide gut microbiome, and for this reason, its striking disappearance in human Westernized populations suggests that changes in recent lifestyle variables might be responsible for the disruption of intestinal microbes possibly coevolving with our body since the evolutionary era of primate host diversification.

## Conclusions

In this study, we expanded the fraction of characterized microbial diversity in the highly unexplored non-human primate metagenome, to enable species- and strain-level comparative genomics analysis of the human and non-human primate microbiome and generate hypotheses on relevant coevolutionary trajectories that shaped the current worldwide structure of the human microbiome. Through the application of strain-level single-sample de novo genome assembly on 203 NHP metagenomic samples, we uncovered over 1000 new SGBs expanding the catalog of microbial species recovered from non-human primates by 77% and improving the mappability of NHP metagenomes by over 600%. These newly assembled genomes contributed to the identification of 760 new genus-level and 265 family-level genome bins that represent completely uncharacterized microbial clades never observed in humans. Compared to the over 150,000 MAGs available from human metagenomes [11] and because of multiple primate hosts that need to be studied, the NHP microbiome still remains undersampled.

Despite the genomes assembled from metagenomes are not free from assembly problems [64, 65] and should be considered for complementing rather than substituting those obtained from isolate sequencing, large-scale metagenomic assembly efforts to mine available metagenomic data showed to be crucial to uncover the whole diversity of environment-specific microbiomes [11, 66, 67], especially in these under-investigated hosts. Indeed, given the efficiency of metagenomic assembly pipelines [67, 68] and the availability of complementary tools to explore the microbial diversity in a microbiome [69, 70], the limiting factor appears to be the technical difficulties in sampling primates in the wild.

The newly established collection of NHP microbial species showed that at the fine-grained taxonomic resolution, there is little overlap between the gut microbiomes of humans and NHPs, with 6% of the overall species found in wild NHP that were identified at least once in human microbiomes. Captive NHPs exposed to more human-like environments and diets showed instead higher species sharing with humans (49%) and a higher degree of metagenome mappability. On the other hand, microbiomes from wild NHPs overlapped comparatively much more (163%) with human populations adopting non-Westernized rather than Westernized lifestyles. Because lifestyle patterns appear to have an impact on the structure of the gut microbiome comparable in effect size to that of the primate host species, NHP and potentially ancient microbiome samples [17–21] are thus more suitable for host-microbe coevolutionary analyses as they are likely less confounded by recent lifestyle changes.

Our strain-level investigations of specific taxonomic clades (Figs. 4 and 5) showed the presence of both species with strains spanning multiple hosts and of sister species associated with different primates. While the former is suggestive of recent inter-host transmission or common acquisition from common sources, the second can be the basis to study microbial evolution or diversification as a consequence of host speciation, especially if phylogenies can be dated using ancient microbiome samples [71] or other time constraints [72]. Our framework can thus be exploited to study inter-host species and zoonotic microbial transmission that is currently mostly limited to specific pathogens of interest [73–78]. The catalog of primate-associated microbial genomes can thus serve as a basis for a better comprehension of the human microbiome in light of recent and ancient cross-primate transmission and environmental acquisition of microbial diversity.

## Methods

### Analyzed datasets

In our meta-analysis, we considered and curated 6 publicly available gut metagenomic datasets (Fig. 1a and Additional file 1: Table S1) spanning 22 non-human primate (NHP) species from 14 different countries in 5 continents (Additional file 2: Figure S1) and metagenomic samples from healthy individuals from 47 datasets included in the curatedMetagenomicData package [79]. In total, our study considers 203 metagenomic samples from the gut of NHPs and 9428 human metagenomes from different body sites.

The non-human primate datasets were retrieved from 4 studies considering wild animals and 2 studies surveying animals in captivity. All but 1 study produced gut metagenomes of 1 single host species. One work [41]

instead analyzed the gut microbiome of 18 species of wild NHPs from 9 countries (Fig. 1a and Additional file 1: Table S1) to test the influence of folivory on its composition and function and highlighted that host phylogeny has a stronger influence than diet. With a similar approach, [30] shotgun sequenced 19 wild western lowland gorillas (*Gorilla gorilla gorilla*) in the Republic of the Congo as part of a 16S rRNA study including sympatric chimpanzees and modern human microbiomes that demonstrated the compositional divergence between the primate clades' microbiome and the seasonal shift in response to changing dietary habits throughout the year. Orkin et al. [40] exposed similar seasonal patterns linked with water and food availability by surveying the microbiome of 20 wild white-faced capuchin monkeys (*Cebus capucinus imitator*) in Costa Rica. Tung et al. [37] instead found that social group membership and networks are good predictors of the taxonomic and functional structure of the gut microbiome by surveying 48 wild baboons (*Papio cynocephalus*) in Kenya. Studies in captivity instead include [38], who sequenced the gut microbiome of 2 red-shanked doucs langurs (*Pygathrix nemaeus*) in captivity that were fed a specific mix of plants to test for the ability of metabarcoding vs metagenomics to identify the plants eaten by the primates from the feces, and [39], who surveyed the change in microbiome composition and function in 20 cynomolgus macaques (*Macaca fascicularis*) fed either a high-fat and low-fiber or a low-fat and high-fiber diet and showed that the first provoked a change toward a more human-like microbiome. Despite the relevance of these 6 works, none of them attempted at reconstructing novel microbial genomes from NHPs.

### Available genomes used as reference
To define known species-level genome bins (kSGBs), we considered the 80,853 annotated genomes (here referred to as reference genomes) available as of March 2018 in the NCBI GenBank database [80]. These comprise both complete (12%) and draft (88%) genomes. Draft genomes include also metagenome-assembled genomes (MAGs) and co-abundance gene groups (CAGs).

### Mapping-based taxonomic analysis
As a preliminary explorative test, taxonomic profiling was performed with MetaPhlAn2 [81] with default parameters. Additional profiling was performed by using the parameter "-t rel_ab_w_read_stats" in order to estimate the read mappability for each profiled species.

### Genome reconstruction and clustering
In order to reconstruct microbial genomes for both characterized and yet-to-be-characterized species, we applied a single-sample metagenomic assembly and contig

binning approach we described and validated elsewhere [11]. Briefly, assemblies were produced with MEGAHIT [42], and contigs longer than 1000 nt were binned with MetaBAT2 [82] to produce 7420 MAGs. Quality control with CheckM 1.0.7 [83] yielded 1033 high-quality MAGs (completeness > 90%, contamination < 5% as described in [11]) and 1952 medium-quality MAGs (completeness > 50% and contamination < 5%). Extensive validation of the MAG reconstruction procedure employed here has been previously validated in [11] by comparing MAGs with isolate genomes obtained from the very same biological sample, including different bacterial species and sample types. This analysis showed that genomes recovered through metagenomic assembly are, at least for the tested cases, almost identical to those obtained with isolate sequencing. Moreover, the specific choices for the use of assemblers, binners, and quality control procedures and of their parameters was proven sound with respect to similar efforts using only partially overlapping methodologies by independent investigations [46].

After metagenomic assembly and binning, MAGs were clustered at 5% genetic distance based on whole-genome nucleotide similarity estimation using Mash (version 2.0; option "-s 10000" for sketching) [84]. Overall, we obtained 99 kSGBs containing at least 1 reference genome retrieved from NCBI GenBank [80], 200 uSGBs lacking a reference genome but clustering together with genomes reconstructed in [11], and 1009 pSGBs consisting of 2186 genomes (73.23% of MAGs recovered from NHPs) newly reconstructed in this study (Fig. 1c). However, even when using a 10% genetic distance to define new SGBs, the ratio of MAGs assigned to pSGBs remained very high with respect to the total MAGs recovered from NHPs (63.75%). SGBs were further clustered into genus-level genome bins (GGBs) and family-level genome bins (FGBs) spanning 15% and 30% genetic distance, respectively.

### Phylogenetic analysis
Phylogenies were reconstructed using the newly developed version of PhyloPhlAn [55]. The phylogenetic trees in Figs. 1b and 4c are based on the 400 universal markers as defined in PhyloPhlAn [55] and have been built using the following set of parameters: "--diversity high --fast --remove_fragmentary_entries --fragmentary_threshold 0.67 --min_num_markers 50 --trim greedy" and "--diversity low --accurate --trim greedy --force_nucleotides," respectively.

From the reconstructed phylogeny in Fig. 1b, we extracted the SGBs falling into the *Treponema* subtree, including also pSGBs. We then applied PhyloPhlAn 2 on all reference genomes and human and non-human primates microbial genomes belonging to the extracted SGBs to produce the phylogenetic tree reported in Fig. 5a

[with params --diversity low --trim greedy --min_num_ marker 50].

External tools with their specific options as used in the PhyloPhlAn framework are as follows:

- diamond (version v0.9.9.110 [85]) with parameters: "blastx --quiet --threads 1 --outfmt 6 --more-sensitive --id 50 --max-hsps 35 -k 0" and with parameters: "blastp --quiet --threads 1 --outfmt 6 --more-sensitive --id 50 --max-hsps 35 -k 0"
- mafft (version v7.310 [86]) with the "--anysymbol" option
- trimal (version 1.2rev59 [87]) with the "-gappyout" option
- FastTree (version 2.1.9 [88]) with "-mlacc 2 -slownni -spr 4 -fastest -mlnni 4 -no2nd -gtr -nt" options
- RAxML (version 8.1.15 [89]) with parameters: "-m PROTCATLG -p 1989"

Trees in Figs. 1b and 5a were visualized with GraPhlAn [90]. The phylogenetic tree of the primates was obtained from [44], manually pruned with iTOL [91] to report only species considered in this study, and visualized with FigTree v.1.4.3 [92].

### Mappability

We estimated the percentage of raw reads in each sample that could align to known bacterial genomes, SGBs, and pSGBs using a previously described method (Pasolli et al. [11]). Briefly, each raw metagenome was subsampled at 1% to reduce the computational cost of mapping. Subsampled reads were filtered to remove alignments to the human genome (hg19). Short (i.e., lower than 70 bp) and low-quality (mean sequencing quality < 20) reads were discarded.

Each sample was mapped against the three groups of indexes: (i) a set of 80,990 reference genomes used to define the set of known SGBs in [11], (ii) the 154,753 known and unknown SGBs from [11], and (iii) the 1009 SGBs from NHPs reconstructed in this study. The mapping was performed with BowTie2 [93] v. 2.3.5 in end-to-end mode. The mapping was performed incrementally (i.e., reads that are reported to map against pSGBs do not map against any reference genome or human SGB). Additionally, BowTie2 alignments scoring less than – 20 (tag AS:i) were excluded, to avoid overestimating the number of mapping reads. The mappability fraction was calculated by dividing the number of aligning reads by the number of high-quality reads within each sample.

### Functional analysis

Metagenome-assembled genomes reconstructed in this study were annotated with Prokka 1.12 [94] using default parameters. Proteins inferred with Prokka were then functionally annotated with UniRef90 and UniRef50 using diamond v0.9.9.110 [85].

KEGG Orthology (KO) for the UniRef50 annotations was retrieved from the UniProt website using the Retrieve/ID mapping tool. KOs related to the metabolism were filtered and used to produce a presence/absence matrix for generating Fig. 5d and Additional file 2: Figure S5. Non-metric multidimensional scaling plots were generated using the Jaccard distance with the metaMDS function in the vegan R package [95].

### Statistical analysis

Statistical significance was verified through Fisher's test with multiple hypothesis testing corrections with either Bonferroni or FDR as reported in the text.

### Supplementary information

---

**Additional file 1: Table S1.** NHP gut metagenome datasets considered in this study, together with relevant information like Pubmed ID, host, wild or captivity status, country of sampling, number of samples.

**Additional file 2: Figure S1.** World map reporting NHP metagenomic samples considered in this study together with host and country information. **Figure S2.** Phylogenetic tree of the Bacteroidetes phylum (uncollapsed version of the tree in Fig. 4b). **Figure S3.** Comparison between the phylogeny of the host species surveyed in this study and the one of FGB 4487, the only FGB spanning three out of the four host clades. Dashed lines link each MAG of the FGB 4487 tree with the host it was retrieved from, thus showing that genetically close hosts tend to harbor genetically similar bacterial strains. **Figure S4.** Phylogenetic tree of the Elusimicrobia phylum (uncollapsed version of the tree in Fig. 4c). **Figure S5.** KO presence/absence profile in *Treponema* MAGs recovered from both stool and oral cavity samples. Only KOs related to metabolism and present in at least 20% and less than 80% of samples are reported.

**Additional file 3: Table S2.** MetaPhlAn2 profiles of all metagenomic samples from NHPs considered in this study.

**Additional file 4: Table S3.** Estimated mapped reads according to MetaPhlAn2, both per sample and per dataset.

**Additional file 5: Table S4.** Description of single-sample assembled genomes (host, wild or captivity status, country of sampling), their assembly and quality statistics, their assigned taxonomy, and SGB, GGB and FGB assignment.

**Additional file 6: Table S5.** KOs associated with NHP and human microbiomes, considering those eight SGBs that are present in at least 10 human and 10 NHP samples. Fisher's test, FDR-corrected $p$-values <0.05.

**Additional file 7: Table S6.** SGB presence/absence in the considered metagenomic samples from NHPs.

**Additional file 8: Table S7.** Description of the 60 kSGBs retrieved from NHPs and shared with humans that are lacking an official species name (e.g. with species name labelled as "sp." or "bacterium"), together with the number of reference genomes and MAGs retrieved for each host category.

**Additional file 9: Table S8.** Description of the SGBs (kSGBs, uSGBs and pSGBs) reported in this study, together with the assigned taxonomy and the number of reference genomes and MAGs retrieved from each host category and NHP dataset.

**Additional file 10: Table S9.** SGBs found in >3 NHP samples and their association with either Westernized or non-Westernized microbiomes

(Fisher's test Bonferroni-corrected p-values, comprehensive list of SGBs shown in Fig. 3a).

**Additional file 11: Table S10. Tab1)** GGBs description, together with number of MAGs assigned and number of dataset and samples in which the GGB is found, divided by host and NHP dataset. **Tab2)** Same information for FGBs recovered in this study.

**Additional file 12: Table S11.** List of *Treponema* SGBs reported in the phylogeny in Fig. 5a and their prevalence in NHPs and humans.

**Additional file 13: Table S12.** Full list of SGBs.

**Additional file 14:** Review history.

## Acknowledgements
The authors would like to acknowledge the Laboratory of Computational Metagenomics for the valuable input.

## Review history
The review history is available as Additional file 14.

## Peer review information

## Authors' contributions
NS and SM conceived and supervised the study. SM, FA, and FB conducted the computational and statistical analysis. SM, FA, FB, EP, and NS analyzed and interpreted the results. EP and DB performed the assembly. FC, FA, FB, NK, and PM performed the clustering. MZ calculated the metagenome mappability. EN performed the Elusimicrobia phylogenetic analysis. MIM performed the explorative MetaPhlAn2 taxonomic profiling. SM and NS wrote the manuscript. All authors read and approved the final manuscript.

## Authors' information
Twitter handles: @BacilluSubtilis (Serena Manara), @epasolli (Edoardo Pasolli), and @nsegata (Nicola Segata).

## Funding
This work was supported by the European Research Council (ERC-STG project MetaPG-716575), MIUR "Futuro in Ricerca" RBFR13EWWI_001, and the European Union (H2020-SFS-2018-1 project MASTER-818368) to NS.

## Availability of data and materials
The non-human primate microbiome datasets analyzed in the current study are available in the Sequence Read Archive (SRA) under BioProjects PRJNA382701 [30], PRJNA271618 [37], and PRJNA485217 [40]; in the European Nucleotide Archive (ENA) under accessions ERP104379 [41] and PRJEB22765 [39]; and in MG-RAST project "Douc Illumina" http://metagenomics.anl.gov/linkin.cgi?project=2562, under IDs 4506440.3 and 4506441.3 (PN1) and 4514928.3 and 4514929.3 (PN2) [38]. All human microbiome datasets are available in the curatedMetagenomicData package [79]. MAGs reconstructed in this study are available in ENA under accession PRJEB35610 [96] and at http://segatalab.cibio.unitn.it/data/Manara_et_al.html. The full list of the SGBs available at the time of the analysis is reported in Additional file 13: Table S12.

## Ethics approval and consent to participate
Not applicable.

## Consent for publication
Not applicable.

## Competing interests
The authors declare that they have no competing interests.

## Author details
¹CIBIO Department, University of Trento, Trento, Italy. ²Department of Agricultural Sciences, University of Naples Federico II, Portici, Italy.

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

## 
