## [**Additional file 14:** Review history. · Genome Biology]

Review History

First round of review

Reviewer 1

Are you able to assess all statistics in the manuscript, including the appropriateness of statistical tests used?

No, I do not feel adequately qualified to assess the statistics.

Comments to author:

This study reconstructed over 1,000 uncharacterized metagenome-assembled-genomes (MAGs) from non-human primates, greatly expanding microbiome genomic resources for primates. The observation that many microbes found in the non-human primate gut microbiome represent new, undescribed species is important. I also think the result that captive non-human primates harbor microbial species and strains more similar to those in humans, as compared to wild primates, is of broad interest because it supports the idea that microbial exposures and host lifestyle play a strong role in shaping primate microbiome composition.

I am relatively new to MAG assembly methods, therefore, this manuscript should be reviewed by someone with extensive expertise in this area. However, I have two comments about the approaches I would like to see addressed.

1. There have been some recent, high-profile critiques of using MAG assemblies to identify new microbial species. These papers argue that some species identified based on MAGs alone are, in fact, artifacts of the assembly process (e.g. see Garg et al. 2019, bioRxiv; doi: <https://doi.org/10.1101/731091> for a recent example). Without culturing, it is difficult to prove whether a given MAG is derived from a single chromosome, or instead represents DNA from multiple chromosomes in the same environment. The authors should address this literature to this paper and describe the steps taken to avoid artifactual identification of new species in their pipelines.

2. Microbial genomes are notoriously flexible, and I wonder how that flexibility plays into the authors' designations of the NHP MAGs into known and unknown species-level genome bins (kSGBs and uSGBs). For instance, if the newly constructed NHP MAGs were subset to only core housekeeping genes (e.g. the 400 universal markers in PhyloPhlAn), does the proportion of MAGs assigned to kSGBs and uSGBs change? In other words, do some NHP MAGs appear to be new species largely because of the flexible, potentially horizontally transmitted components of their genome? Or alternatively, does the percentage of uSGB designations stay the same when the MAGs are subset to only vertically inherited housekeeping genes?

Minor comments

Bottom of page 3 - can you provide percentages for UniRef50 assignments per MAG, in addition to the raw number of assignments?

Page 6, lines 28-32. Consider citing Clayton et al 2016 PNAS, which finds that captivity has humanizing effects on primate microbiota.

The manuscript contains several small grammatical errors and typos. Below are a few such issues, but I did not record all of them.

page 11 line 55. Please specify that the studies produced metagenomes of one host species, to avoid potential confusion as to whether the sentence refers to host or microbial species.

Page 12, lines 7 and 10: check capitalization on sentences that start with reference numbers.

Page 12 line 41: The word "Genomes" should not be plural in this sub-heading

Reviewer 2

Are you able to assess all statistics in the manuscript, including the appropriateness of statistical tests used?

Yes, and I have assessed the statistics in my report.

Comments to author:

This study presents the assembly and analysis of nearly 3000 partial genomes from the gut microbiomes of non-human primates. The data produced will be extremely valuable for future metagenomics studies of humans and NHPs. The results of analyses, including the observations that captivity partially humanizes the primate gut microbiome, that humans have lost microbial diversity, and that westernized peoples harbor less microbiota overlap with nonhuman primates than do non-westernized peoples, largely confirm previous studies. The major advance of this study is the assembly of many new NHP gut bacterial genomes, which is significant/important.

It would be interesting to include more information about functional differences between human and NHP microbiota. Do gut bacterial genomes from NHPs differ consistently in gene content from those of humans? Such an analysis could produce new insights/hypotheses about processes driving differences between humans and NHPs, which would go beyond describing genomes. The authors report some differences in ability to functionally classify genes between humans and NHPs, but perhaps there is a way to account for this.

The authors use the term coevolution throughout, but it is perhaps not the correct term in most places, eg, line 56. Coevolution refers specifically to reciprocal adaptations between lineages, which may or may not occur in cases where lineages codiversify. The results showing phylogenetic distinctiveness of strains from humans vs strains from NHPs provide evidence for codiversification, but don't necessarily speak to whether particular microbes coevolve with hosts.

The authors note that their results suggest the possibility of co-diversification/co-speciation. Note that previous studies have reported this process in apes and other mammals (eg, <https://science.sciencemag.org/content/353/6297/380> and <https://www.nature.com/articles/ncomms14319>)

Figure 4: Is it possible to conduct an analysis that tests for congruent phylogenetic histories between hosts and bacteria? (eg <https://www.sciencedirect.com/science/article/pii/S2214574516301559>) Such an analysis could provide a better test of whether or not bacteria are codiversifying with humans and NHPs.

How does geography affect the patterns of strains sharing between host species (while controlling for host phylogenetic history)? Do unrelated NHPs living in geographic proximity harbor more of the same strains than when the hosts live apart? This may be outside the scope of the study, but would be interesting to look at. Similarly, it would be interesting to test whether NHPs and humans living in the same region share more strains than hosts from different regions.

It would also be useful to dig more deeply into the effect of captivity on strain sharing between humans and NHPs. Are there any particular bacterial taxa that tend to be shared or not to be shared? It may be that some bacterial taxa display host specificity to NHPs despite captivity, which could suggest there exist some mechanisms ensuring their stability.

Reviewer #1

This study reconstructed over 1,000 uncharacterized metagenome-assembled-genomes (MAGs) from non-human primates, greatly expanding microbiome genomic resources for primates. The observation that many microbes found in the non-human primate gut microbiome represent new, undescribed species is important. I also think the result that captive non-human primates harbor microbial species and strains more similar to those in humans, as compared to wild primates, is of broad interest because it supports the idea that microbial exposures and host lifestyle play a strong role in shaping primate microbiome composition.

Response.

We thank the reviewer for the comment, and we are happy that our work was appreciated.

I am relatively new to MAG assembly methods, therefore, this manuscript should be reviewed by someone with extensive expertise in this area. However, I have two comments about the approaches I would like to see addressed.

1. There have been some recent, high-profile critiques of using MAG assemblies to identify new microbial species. These papers argue that some species identified based on MAGs alone are, in fact, artifacts of the assembly process (e.g. see Garg et al. 2019, bioRxiv; doi: <https://doi.org/10.1101/731091> for a recent example). Without culturing, it is difficult to prove whether a given MAG is derived from a single chromosome, or instead represents DNA from multiple chromosomes in the same environment. The authors should address this literature to this paper and describe the steps taken to avoid artifactual identification of new species in their pipelines.

Response.

We thank the reviewer for this comment. While we can demonstrate the high quality of our MAGs with tools complementary to the standard QC measure employed for MAGs (below), it is not our intention to suggest that MAGs are of the same quality of genomes from isolate sequencing. Yet, our point is that even using genomic information that is not as accurate as isolate sequencing, we are able to obtain sound results on the overall diversity and phylogenetic structure of the gut microbiome in NHPs. In the revised manuscript, we thus (1)

Department of Cellular, Computational and Integrative Biology - CIBIO

provide more details on the quality control applied on the MAGs (that we initially failed to include in the original manuscript) and (2) underline that MAGs are essential in the analysis of microbiomes (as comparable number of genomes from isolate sequencing is unfeasible) but are not as reliable as isolate sequencing.

On the discussion of the quality of the genomes, what we have missed to write in the paper, is that we used the same exact procedure we validated in the Pasolli et al work (Pasolli et al, Cell, 2019, DOI:<https://doi.org/10.1016/j.cell.2019.01.001>) using several examples of combined sequencing of microbiomes and isolate genomes. In the Pasolli et al paper we took a comprehensive approach at evaluating the quality of MAGs that applies in the same way to human and NHP microbiomes. In particular, we compared reconstructed MAGs with isolate genomes retrieved from the very same sample (or from samples obtained from the same individual at different times), for eight different bacterial species retrieved from different sample types, as reported in Figures 7A-C of the paper (reported below for convenience). This allowed us to show that genomes reconstructed from metagenomes were almost identical to the ones from isolate sequencing (see STAR Methods and Figures 7A-C of the Pasolli et al paper).

While this analysis cannot be performed on thousands of cases, it clearly supports the statement that, after strict quality control, MAGs are reliable sources of strain-level genetic information. Moreover, new results combining multiple large-scale metagenomic assembly approaches (Almeida, Nature, 2019; Sberro et al, Cell, 2019) showed that the main results are independently confirmed across groups using only partially overlapping methodologies (Almeida, bioRxiv, 2019).

We clarify that the MAG construction and validation pipeline has already been extensively validated as follows in the results:

“We thus employed an assembly-based approach we previously proposed and validated elsewhere (Pasolli et al. 2019) (see Methods) and that was also recently cross-checked with other similar efforts (A. Almeida et al. 2019) to reconstruct microbial genomes de novo in the whole set of available NHP metagenomic samples.”

and in the methods:

“Extensive validation of the MAG reconstruction procedure employed here has been previously validated in (Pasolli et al. 2019) by comparing MAGs with isolate genomes obtained from the very same biological sample, including different bacterial species and sample types. This analysis showed that genomes recovered through metagenomic assembly are, at least for the tested cases, almost identical to those obtained with isolate sequencing. Moreover, the specific choices for the use of assemblers, bidders, and quality control procedures and of their parameters was proven sound with respect to similar efforts using only partially overlapping methodologies by independent investigations (A. Almeida et al. 2019).”

We then stressed in the text (also pointing at the relevant literature) that MAGs are crucial to uncover the diversity of the microbiome and to drive metagenomic analysis, despite the fact that they cannot be considered on the same level of genomes from isolate sequencing.

“Compared to the over 150,000 MAGs available from human metagenomes (Pasolli et al. 2019) and because of multiple primate hosts that need to be studied, the NHP microbiome still remains undersampled.

Despite genomes assembled from metagenomes are not free from assembly problems (Garg et al. 2019; Chen et al. 2019) and should be considered for complementing rather than substituting those obtained from isolate sequencing, large-scale metagenomic assembly efforts to mine available metagenomic data showed to be crucial to uncover the whole diversity of environment-specific microbiomes (Quince et al. 2017; Nayfach et al. 2019; Pasolli et al. 2019), especially in these under-investigated hosts. Indeed, given the efficiency of metagenomic assembly pipelines (Nayfach et al. 2019; Alexandre Almeida et al. 2019) and the availability of complementary tools to explore the microbial diversity in a microbiome (Zou et al. 2019; Gawad, Koh, and Quake 2016), the limiting factor appears to be the technical difficulties in sampling primates in the wild.”

2. Microbial genomes are notoriously flexible, and I wonder how that flexibility plays into the authors' designations of the NHP MAGs into known and unknown species-level genome bins (kSGBs and uSGBs). For instance, if the newly constructed NHP MAGs were subset to only core housekeeping genes (e.g. the 400 universal markers in PhyloPhlAn), does the proportion of MAGs assigned to kSGBs and uSGBs change? In other words, do some NHP MAGs appear to be new species largely because of the flexible, potentially horizontally transmitted components of their genome? Or alternatively, does the percentage of uSGB designations stay the same when the MAGs are subset to only vertically inherited housekeeping genes?

Response.

We thank the Reviewer for raising this question, which was not extensively explained. Also in this case, the methodology we employed to define species-level genome bins, was previously proposed and validated in the work by Pasolli et al (Pasolli et al, Cell, 2019). In the Pasolli work, the 5% genetic diameter on SGB definition was selected as the one that better recapitulates taxonomically labeled species-level clusters in 61,198 reference genomes spanning 5,494 taxonomically assigned species in NCBI GenBank (please see STAR Methods and Figure S2 of Pasolli paper; Figure S2 is reported below for convenience). As reported in the Pasolli paper, also an independent work by Jain et al (Jain et al, Nat. Commun., 2018) proposed the same range to define prokaryotic species. This threshold was also verified empirically (see Figure S2C below) by evaluating how well this

Department of Cellular, Computational and Integrative Biology - CIBIO

approach can recapitulate existing species, and we indeed found and confirmed Konstantinidis and Varghese results (Konstantinidis et al, PNAS, 2004; Varghese et al, NAR, 2015), showing that 5% ANI (or MASH) distance captures the intended variability in the core genome genetics irrespective of rearrangements and genomic variability due to horizontal gene transfer.

However, even if we changed the threshold to define new SGBs from 5% to 10% mash distance, therefore allowing for a larger intra-species genomic variability and therefore flexibility, the number of MAGs assigned to pSGBs would change from 2,186 to 1,903, with 283 MAGs previously assigned to pSGBs being assigned to kSGBs and uSGBs (109 and 174 MAGs, respectively).

MAGs assigned to each SGB type		
SGB type	5% threshold	10% threshold
kSGB	310	419
uSGB	489	663
pSGB	2186	1903

We added a brief comment on this in the Methods, as follows:

“Overall, we obtained 99 kSGBs containing at least one reference genome retrieved from NCBI GenBank (NCBI Resource Coordinators 2018), 200 uSGBs lacking a reference genome but clustering together with genomes reconstructed in (Pasolli et al. 2019), and 1,009 pSGBs consisting of 2,186 genomes (73.23% of MAGs recovered from NHPs) newly reconstructed in this study (Figure 1C). However, even when using a 10% genetic distance to define new SGBs, the ratio of MAGs assigned

Department of Cellular, Computational and Integrative Biology - CIBIO

to pSGBs remained very high with respect to the total MAGs recovered from NHPs (63.75%).”

and in the Results as follows:

“The large majority of the MAGs remained however unassigned, with 2,186 MAGs (73.23%) showing >5% genetic distance to any SGB, and 1,903 MAGs (63.75%) showing >10% genetic distance.”

Figure S2. Overview of the Reconstructed SGBs and Criteria for SGB Definition and Taxonomic Assignment, Related to Figure 6. (A) Distribution of the distances of each reconstructed genome to the closest available isolate genomes, grouped by the class assigned to the matching isolate genomes. (B) The 4,930 identified species-level genome bins (SGBs) comprise a very variable fraction of already available genomes versus genomes we reconstructed from metagenomes. (C) Minimization criterion adopted to find the optimal cutoff in the hierarchical clustering of genomes to define SGBs. Two criteria are taken into account: minimization of the over-clustering error (C-i), and minimization of the under-clustering error (C-ii). Results showed a minimization of the error for a threshold equal to 0.05 (C-iii), which was thus adopted to discretize subtrees in the dendrogram and generate SGBs spanning ~5% genetic diversity. (D) The same minimization criterion reported in (C-iii) for species-level bins is also adopted to identify the genomic diversity for genus-level and family-level bins.

Minor comments

- **Bottom of page 3 - can you provide percentages for UniRef50 assignments per MAG, in addition to the raw number of assignments?**

Response.

We agree with the Reviewer that the ratio of annotated/predicted proteins gives a better idea of the low level of functional characterization of NHP's MAGs. We therefore modified the text as follows:

“Functional annotation of all MAGs (see Methods, (“UniProt: The Universal Protein Knowledgebase” 2016)) showed low levels of functional characterization in NHPs, with only 1,049±482 UniRef50 (61.9%±17.3% st.dev. of predicted proteins) assigned per MAG, in contrast with the 1,426±591 (77.3%±14.6% st.dev. of predicted proteins) assigned to MAGs from non-Westernized human samples and 1,840±847 (83.7%±12.6% st.dev. of predicted proteins) assigned to those obtained from Westernized human populations.”

- **Page 6, lines 28-32. Consider citing Clayton et al 2016 PNAS, which finds that captivity has humanizing effects on primate microbiota.**

Response.

We thank the Reviewer for the suggestion. We added the citation as follows:

“However, both the two most represented human kSGBs assigned to the *Prevotella* genus (13 and 11 MAGs recovered respectively, Figure 2A and Additional file 9: Table S8), were retrieved from *Macaca fascicularis* in captivity from the LiX_2018 dataset, consistently with previous literature (Amato et al. 2015; Ma et al. 2014; Clayton et al. 2016).”

- **The manuscript contains several small grammatical errors and typos. Below are a few such issues, but I did not record all of them.**
 - **page 11 line 55. Please specify that the studies produced metagenomes of one host species, to avoid potential confusion as to whether the sentence refers to host or microbial species.**
 - **Page 12, lines 7 and 10: check capitalization on sentences that start with reference numbers.**
 - **Page 12 line 41: The word "Genomes" should not be plural in this sub-heading**

Response.

We thank the Reviewer for catching these typos and errors. We thoroughly checked the text for grammatical and spelling errors that are highlighted in the Track Changed version of the work.

Reviewer #2

This study presents the assembly and analysis of nearly 3000 partial genomes from the gut microbiomes of non-human primates. The data produced will be extremely valuable for future metagenomics studies of humans and NHPs. The results of analyses, including the observations that captivity partially humanizes the primate gut microbiome, that humans have lost microbial diversity, and that westernized peoples harbor less microbiota overlap with nonhuman primates than do non-westernized peoples, largely confirm previous studies. The major advance of this study is the assembly of many new NHP gut bacterial genomes, which is significant/important.

Response.

We thank the reviewer for the positive comment on our work.

It would be interesting to include more information about functional differences between human and NHP microbiota. Do gut bacterial genomes from NHPs differ consistently in gene content from those of humans? Such an analysis could produce new insights/hypotheses about processes driving differences between humans and NHPs, which would go beyond describing genomes. The authors report some differences in ability to functionally classify genes between humans and NHPs, but perhaps there is a way to account for this.

Response.

We thank the Reviewer for the suggestion. Because the gene content of NHP microbiomes is substantially less annotated than that of the human microbiome, and because accurate functional comparison between very different taxa is difficult to interpret, we chose to focus on differences between related taxa in NHP and humans. We study what functional diversification could have occurred in related taxa in the two host types, we selected those SGBs that were found in at least 10 NHP and 10 human microbiomes. Due to the low level of overlap between human and NHP bacterial species, only eight SGBs satisfied these requirements, therefore providing perhaps limited generalizability. Statistical analysis with FDR correction on the functional annotations of these eight SGBs showed 150 KOs significantly associated with NHP strains and 166 KOs that were instead associated with human strains, that we briefly commented in the results.

We added the following paragraph in relevant section of the Results:

“Comparative functional analysis between human and NHP strains was hindered by the low level of overlap between species-level genome bins (SGBs, i.e. clusters of MAGs spanning 5% genetic diversity, see Methods) retrieved from human and NHP metagenomes, with only eight SGBs being present in at least ten human and ten NHP microbiomes. Statistical analysis on the functional annotations of these shared SGBs showed 150 KOs (KEGG Orthology) significantly associated with NHP strains and 166 KOs associated with human strains (Fisher’s test FDR-corrected p-values <0.05, Additional File 6: Table S5). Among functions associated with NHP strains, we found different genes involved in the degradation of sugars like cellobiose (K00702, K02761) and maltose (K16211, K01232), and among those associated with human ones, genes encoding for the degradation of different antibiotic compounds, including penicillin and vancomycin (K01710, K02563, K07260, K07259), which is consistent with the exposure of humans but not NHPs to antibiotics.”

This analysis focused on the host-specific putative adaptation in species shared between the hosts. As mentioned, a larger analysis irrespective of phylogeny is hampered by the very low annotation rate of NHP MAGs and we prefer not to include it here.

The authors use the term coevolution throughout, but it is perhaps not the correct term in most places, eg, line 56. Coevolution refers specifically to reciprocal adaptations between lineages, which may or may not occur in cases where lineages codiversify. The results showing phylogenetic distinctiveness of strains from humans vs strains from NHPs provide evidence for codiversification, but don’t necessarily speak to whether particular microbes coevolve with hosts.

Response.

We agree with the Reviewer, and we substituted “coevolution” with “co-diversification” in multiple instances that are reported below:

Line 32: “Conclusions. The newly reconstructed species greatly expand the microbial diversity associated with NHPs, thus enabling better interrogation of the primate

microbiome and empowering in-depth human and non-human comparative and co-diversification studies.”

Line 59: “Some patterns of co-diversification between humans and their microbiomes can be in principle investigated by comparative and phylogenetic analysis of genomes and metagenomes in non-human primates (NHPs), the closest evolutionary relatives of humans (Amato 2019).”

Line 74: “Yet, because this approach is low-resolution and lacks functional characterization, many co-diversification aspects cannot be studied.”

Line 80: “However, because of the advances in metagenomic assembly (D. Li et al. 2015; Nurk et al. 2017) and its application on large cohorts (Pasolli et al. 2019), there is now the possibility to compile a more complete catalog of species and genomes in NHP microbiomes and thus enable accurate co-diversification and comparative analyses.”

Line 349: “This likely reflects a complex evolutionary pattern in which vertical co-diversification (Moeller, Caro-Quintero, et al. 2016; Cruaud and Rasplus 2016), independent niche selection, and between-host species transmission are likely all simultaneously shaping the members of the gut microbiome of primates.”

The authors note that their results suggest the possibility of co-diversification/co-speciation. Note that previous studies have reported this process in apes and other mammals (eg, <https://science.sciencemag.org/content/353/6297/380> and <https://www.nature.com/articles/ncomms14319>)

Response.

We thank the Reviewer for suggesting these works, which were added in the text as follows:

“This likely reflects a complex evolutionary pattern in which vertical co-diversification (Moeller, Caro-Quintero, et al. 2016; Cruaud and Rasplus 2016), independent niche selection, and between-host species transmission are likely all simultaneously shaping the members of the gut microbiome of primates.”

Figure 4: Is it possible to conduct an analysis that tests for congruent phylogenetic histories between hosts and bacteria? (eg <https://www.sciencedirect.com/science/article/pii/S2214574516301559>) Such an analysis could provide a better test of whether or not bacteria are codiversifying with humans and NHPs.

Response.

To test the hypothesis suggested by the Reviewer, we selected the taxonomically unassigned FGB 4487, which is the one spanning the largest host diversity, covering three of the four main host clades (Lemuriformes, Platyrrhini, Cercopithecoidea, but no Hominoidea). The 15 MAGs assigned to this FGB were reconstructed from 7 wild host species from 6 countries. The phylogeny of FGB 4487 recapitulated the phylogeny of the host species, showing specific subtrees for the Lemuriformes, Platyrrhini, and Cercopithecoidea clades.

We included this analysis in our work, as Additional File 2: Figure S3 (see below) and in the text as follows:

“To further investigate the hypothesis of at least occasional paired primate-microbe co-diversification, we selected the taxonomically unassigned FGB 4487, which is the only FGB retrieved in this work that spans three out of the four main host clades (Lemuriformes, Platyrrhini, Cercopithecoidea, but no Hominoidea), including 15 MAGs reconstructed from seven wild hosts from six countries. The phylogeny of FGB 4487 recapitulated the one of the hosts (Additional file 2: Figure S3), with different same-clade host species from different countries sharing the same SGB (e.g. different *Alouatta* species from three different countries sharing pSGB 20386) while being distinct from those found in other clades, thus supporting the hypothesis that host-microbiome co-diversification could have occurred at least for some bacterial clades.”

This new analysis indeed provide sequence-level evidence of potential co-diversifying microbes within the primate evolution.

How does geography affect the patterns of strains sharing between host species (while controlling for host phylogenetic history)? Do unrelated NPHs living in geographic proximity harbor more of the same strains than when the hosts live apart? This may be outside the scope of the study, but would be interesting to look at. Similarly, it would be interesting to test whether NHPs and humans living in the same region share more strains than hosts from different regions.

Response.

We thank the Reviewer for the suggestion. We agree it would be an interesting topic to investigate, but because of the low number of samples per single host and country (n=8 at maximum), our analysis is very unlikely to produce statistically supported results. However, we decided to briefly test this hypothesis and got results suggesting that hosts living in close geographic proximity are more likely to share the same SGB than hosts living in different countries. Focusing on the AmatoKR_2018 dataset and considering only SGBs found in at least two different host/country combinations, we indeed found that each SGB was shared on average 1.7% between hosts belonging to different countries and 4.44% between

Department of Cellular, Computational and Integrative Biology - CIBIO

different hosts belonging to the same country. This hypothesis should however be further investigated in larger NHP datasets and we prefer not proposing this hypothesis in the present work.

It would also be useful to dig more deeply into the effect of captivity on strain sharing between humans and NHPs. Are there any particular bacterial taxa that tend to be shared or not to be shared? It may be that some bacterial taxa display host specificity to NHPs despite captivity, which could suggest there exist some mechanisms ensuring their stability.

Response.

This is an interesting suggestion. There are 778 SGBs that are uniquely found in wild NHPs (18 kSGBs and 18 uSGBs previously found in human gut metagenomes, and 742 pSGBs never found in human gut metagenomes) and 506 that are found only in captive NHPs (71 kSGBs and 174 uSGBs previously found in human gut metagenomes, and 261 pSGBs never found in human gut metagenomes). Overall, only 24 SGBs are shared between wild and captive NHPs (10 kSGBs, 8 uSGBs, and 6 pSGBs).

Focusing on these SGBs consistently found in both wild and captive NHPs, six pSGBs (by definition, never found in human microbiomes before) are highly uncharacterized, with three of them assigned to the Acetobacteraceae (SGB ID 20294, n=5), Clostridiaceae (SGB ID 20600, n=4), and Prevotellaceae (SGB ID 20101, n=4) families, and the other three only assigned at the phylum level (Tenericutes (SGB ID 21081, n=5), Firmicutes (SGB ID 20681, n=3), and Bacteroidetes (SGB ID 20197, n=2)). These might be bacterial taxa that are inherently linked with NHPs gut environments and are not lost despite captivity.

The remaining 18 kSGBs and uSGBs present both in captive and wild NHPs and shared with humans mainly belong to unclassified Firmicutes (n=5) and uncharacterized *Ruminococcus* species (n=4). Among the most prevalent in NHPs, the kSGBs of *Treponema berlinense*, *Succinatimonas* sp., *Escherichia coli*, and *Prevotella* sp. These species are likely key players in the primate gut microbiome, as they are consistently found in different host species independently from captive or wild NHP lifestyle.

We include part of these information in the text as follows:

“Nevertheless, a few SGBs were consistently found in both wild and captive NHPs and shared with humans. These 10 kSGBs and 8 uSGBs mainly belonged to unclassified Firmicutes (n=5) and uncharacterized *Ruminococcus* species (n=4). Among the most prevalent in NHPs, the kSGBs of *Treponema berlinense*, *Succinatimonas* sp., *Escherichia coli*, and *Prevotella* sp. were consistently found in different host species spanning NHPs and humans and thus appear as key players in the primate gut microbiome.”

Second round of review

Reviewer 1

This study makes important contribution to the literature, and I am now full satisfied with the paper. The revisions adequately address my concerns about the quality of the MAG assemblies and SGB definitions. I have no further suggestions to improve this paper.

Reviewer 2

The authors have addressed all major comments.